# MiraGe: Editable 2D Images using Gaussian Splatting

## Abstract

Implicit Neural Representations (INRs) approximate discrete data through continuous functions and are commonly used for encoding 2D images. Traditional image-based INRs employ neural networks to map pixel coordinates to RGB values, capturing shapes, colors, and textures within the network's weights. Recently, GaussianImage has been proposed as an alternative, using Gaussian functions instead of neural networks to achieve comparable quality and compression. Such a solution obtains a quality and compression ratio similar to classical INR models but does not allow image modification. In contrast, our work introduces a novel method, MiraGe, which uses mirror reflections to perceive 2D images in 3D space and employs flat-controlled Gaussians for precise 2D image editing. Our approach improves the rendering quality and allows realistic image modifications, including human-inspired perception of photos in the 3D world. Thanks to modeling images in 3D space, we obtain the illusion of 3D-based modification in 2D images. We also show that our Gaussian representation can be easily combined with a physics engine to produce physics-based modification of 2D images. Consequently, MiraGe allows for better quality than the standard approach and natural modification of 2D images.

## 1 Introduction

Recent research has increasingly emphasized human perception and the understanding of the world through this lens (Lu, 2019; Davoodi et al., 2023). In line with this trend, we introduce a model that encodes 2D images by simulating human interpretation. Specifically, our model perceives a 2D image as a human would view a photograph or a sheet of paper, treating it as a flat object within a 3D space. This approach allows for intuitive and flexible image editing, capturing the nuances of human perception while enabling complex transformations (see Fig. 1).

Gaussian Splatting (3DGS) framework models the structure of a 3D scene using Gaussian components (Kerbl et al., 2023). In the 2D domain, GaussianImage (Zhang et al., 2024) has shown promising results in image reconstruction by efficiently encoding images in the 2D space, with a strong focus on model efficiency and reduced training time. Unfortunately, GaussianImage does not support user-driven adjustments of scene objects, which is a key feature of 3DGS. While GaussianImage has explored image representation using 2D Gaussians primarily for data compression, our research high-

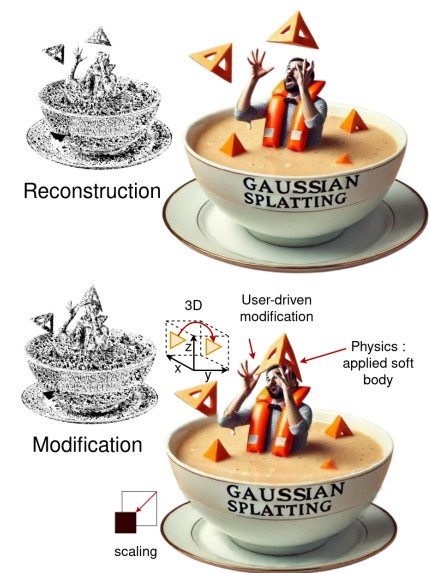

Figure 1: MiraGe encodes 2D images with parameterized Gaussians, enabling high-quality reconstruction and real-life-like modifications. Selected part of image can be transformed in 3D space, creating a 3D effect, with a physics engine controlling movement and interactions.

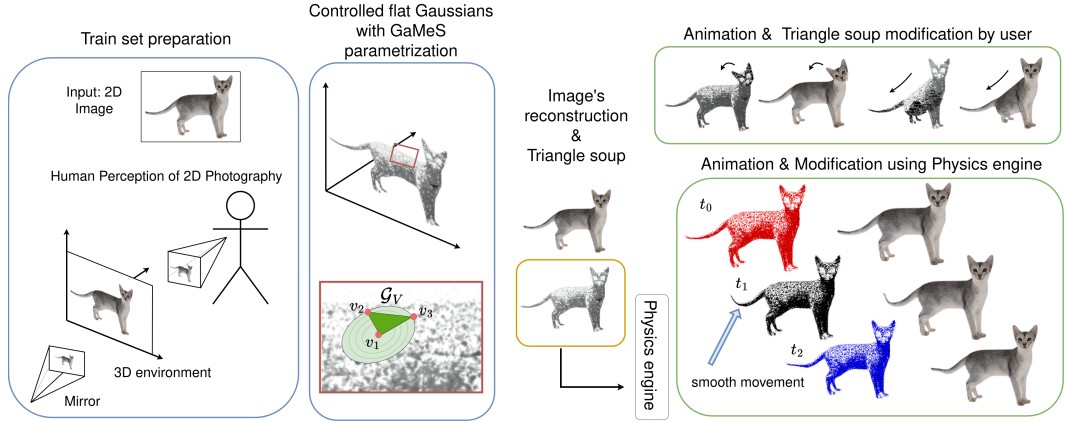

Figure 2: MiraGe employs 3D flat parameterized Gaussians in 3D space to encode 2D images, representing each flat Gaussian as three points, forming a cloud of triangles called a triangle soup. This representation enables real-time manipulation of the 3D triangle/point clouds, allowing for flexible, real-world modifications. The model seamlessly integrates with a physics engine, enhancing its applicability in dynamic environments.

lights an additional benefit, i.e., the use of parameterized flat 3D Gaussians for editing 2D images. In our work, we address this by introducing the MiraGe model, which encodes 2D images through the lens of human perception, bridging the gap between 2D image representation and 3D spatial understanding (see Fig. 2).

Building on the foundational idea that humans intuitively can perform transformations on photographs–primarily through affine transformations and bending them beyond the 2D plane–we introduce a novel approach using flat Gaussians with GaMeS parameterization (Waczyńska et al., 2024b). This capability enables our model to support image editing in both 2D and 3D spaces. Notably, our framework simplifies often difficult perspective adjustments by allowing intuitive modifications directly within the third dimension (see Fig. 3).

In addition to classical edits, our model has the unique capability of interfacing with physics engines, enabling applications that enhance the realism and immersiveness of animations (Jiang et al., 2024). MiraGe treats the physics engine as a black box and offers three distinct methods for controlling Gaussians, i.e., 2D, Amorphous and Graphite. For 2D representation (2D-MiraGe) we used Taichi_elements[1], for 3D representation (Amorphous-MiraGe, Graphite-MiraGe) we use Blender[2]. This flexibility makes our model highly applicable to various fields, such as computer graphics for populating spatial interfaces, where realistic, physics-factual 2D animations can be incorporated (Tadeja et al., 2023).

Embedding 2D images in 3D space allows for seamless integration of 2D and 3D objects, enabling the creation of dynamic backgrounds or interactive elements within animated scenes. This versatility extends to applications such as virtual reality, where 2D images can function as backdrops (Yin et al., 2024). This capability opens up new avenues for creative composition, offering a powerful toolset for users. The novelty of this work lies

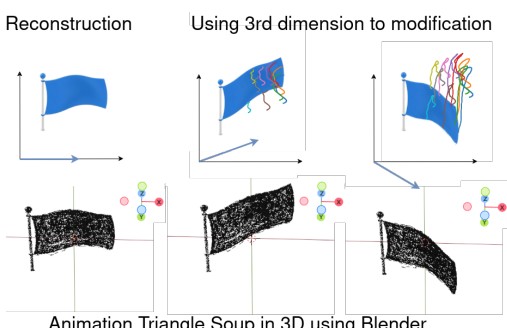

Figure 3: Parameterized flat 3D Gaussians provide a powerful representation of 2D images, enabling flexible editing in 3D space. Triangle Soup can be animated using tools like Blender. The colored lines depict the motion paths of 10 randomly selected points during the simulation.

---

[1] https://github.com/taichi-dev/taichi_elements
[2] https://www.blender.orgversion3.6

in its ability to enable easy, intuitive 3D transformations and integrations within a traditionally 2D framework, expanding the possibilities for both image editing and animation (see Fig. 4).

Since high-quality image reconstruction is critical in animation, we compared MiraGe with other models in particular with GaussianImage (see Fig. 5), showing our state-of-the-art performance in the image reconstruction task. Our results demonstrate that the model operates in real-time, though this comes with a trade-off in terms of model size.

It is worth highlighting that flat 3D Gaussians can be utilized for 2D images in four distinct scenarios, with modifications that emphasize how controlling the Gaussians during training affects the perspective of viewing each image (see Fig. 6).

The following constitutes a list of our key contributions:

- We introduce the MiraGe model, which represents 2D images using flat 3D Gaussian components, achieving state-of-the-art reconstruction quality.

- MiraGe enables the manipulation of 2D images within 3D space, creating the illusion of 3D effects.

- We integrate MiraGe with a physics engine, enabling physics-based modifications and interactions for both 2D and 3D environments.

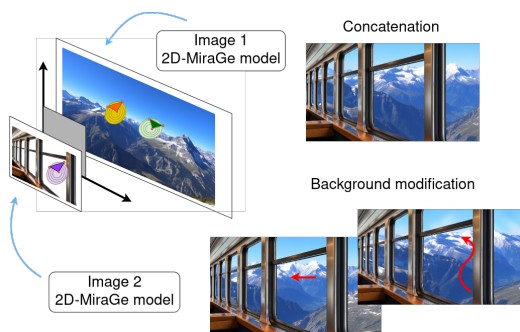

Figure 4: Two images were encoded using the MiraGe model on distinct planes within a 3D space. This setup allows for seamless integration of the encoded images, resulting in a collage-like composition. Moreover, the model facilitates editing capabilities, as illustrated here with modifications to the background image (the rear plane).

## 2 RELATED WORKS

Our work builds on several key research areas, including image reconstruction techniques, Gaussian-based representations and Gaussian animation frameworks.

One rapidly growing area in image reconstruction is Implicit Neural Representations (INRs), which have attracted significant attention for their ability to model continuous signals, such as images, through neural networks (Klocek et al., 2019). INRs encode spatial coordinates and map them to corresponding values, such as RGB color, allowing for highly compact and efficient representations (Xie et al., 2022). This has led to the development of several special-

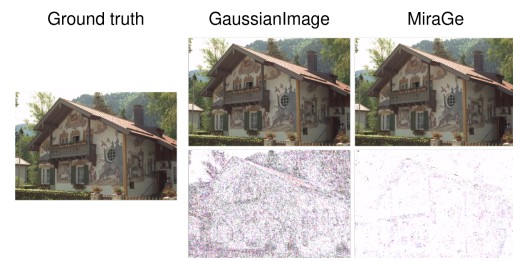

Figure 5: Visual comparison of two Gaussian-based methods for 2D image reconstruction. From left to right, the columns display the ground truth image, the GaussianImage reconstruction, and the MiraGe reconstruction. The bottom row illustrates the differences between the ground truth image and the results of each method.

ized models for image INR, such as SIREN (Sitzmann et al., 2020a) with the novelty of sine used as a periodic activation function to tackle the problem of complex image signals. Fourier feature mapping was proposed in (Tancik et al., 2020) as another answer to the difficulty of aligning the multilayer perceptron (MLP) predictions with high-frequency pictures. Interestingly, authors of WIRE (Saragadam et al., 2023a) have leveraged continuous complex Gabor wavelets to capture visual signals with decent quality. The growing field of research resulted in further improvements of already existing solutions, e.g. in (Liu et al., 2024), certain limitations of SIREN, namely the arising capacity-convergence gap, were successfully alleviated with the idea of variable-periodic activation functions. Yet another worth noting work from this area is (Müller et al., 2022) with INR solution designed to effectively perform on modern computer architecture utilizing a simple data structure concept of hashmap to offer speed-oriented image representation with high fidelity of the outcomes.

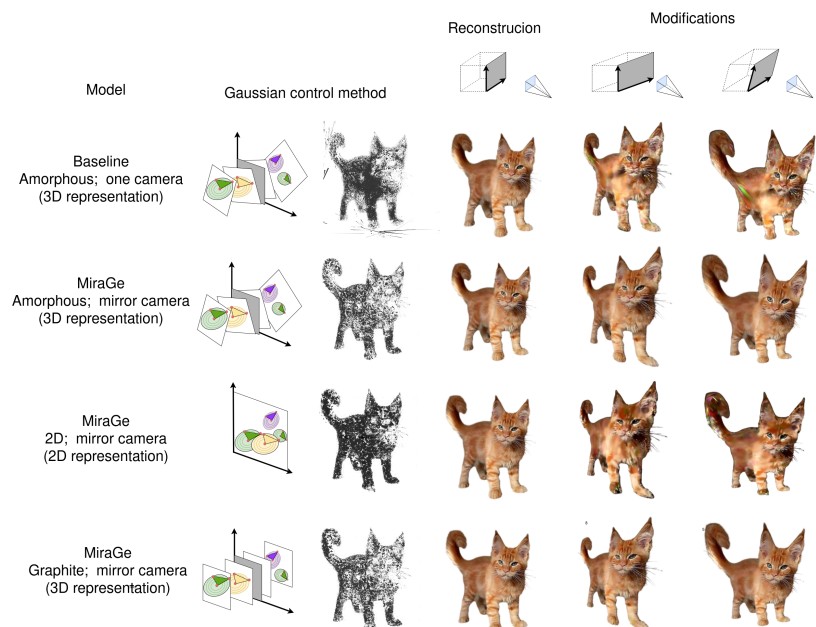

Figure 6: We demonstrate three approaches for Gaussian control: Amorphous, 2D, and Graphite. As a baseline, we utilize a single camera from the Amorphous setup. After applying perspective editing in 3D, the image shows noticeable deformation. In contrast, no deformation is observed when employing the Amorphous or Graphite methods with an additional camera. The model employs a mirror setup during training, with the Amorphous configuration achieving the best results for image reconstruction and 3D analysis. The 2D model represents images on a single plane, allowing 2D physics engines like Taichi_elements to be used, but it does not support 3D modifications. The Graphite model operates across multiple planes, making it ideal for 3D spatial reasoning and image combination.

Alternative approaches to INRs were presented in GaussianImage (Zhang et al., 2024). Instead of neural networks, the authors propose to approximate 2D images using 2D Gaussian components. In practice, GaussianImage is a 2D version of 3DGS (Kerbl et al., 2023) that uses 2D Gaussians instead of their 3D version and a simplified rendering procedure. Thanks to such a modification, the GaussianImage is invariant to the order of Gaussian components. Therefore, such a model is numerically efficient.

GaussianImage represents each pixel color as a weighted sum of 2D Gaussians. The training procedure is similar to 3DGS without pruning. The authors show that such representation gives a similar reconstruction quality to classical INR models and is able to obtain a high compression ratio and fast rendering.

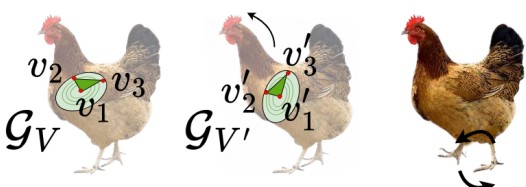

Figure 7: MiraGe use GaMeS (Waczyńska et al., 2024b) representations of flat Gaussian by triangle soup. Therefore, we can use real-life modification by moving points.

The interactive image editing of 2D images has been widely explored in computer graphics. Here, some methods leverage the current advancements in generative models. For instance, Pan et al. (2023) introduce DragGAN, enabling point-based manipulation of images by performing them on the underlying manifold of GAN, achieving realistic edits. Similarly, Shi et al. (2023) propose DragDiffusion, which extends the previous framework to diffusion models, enhancing the control and applicability of image editing. On the other hand, Jacobson et al. (2011) propose bounded biharmonic weights for linear blending, which produce smooth and intuitive deformation for handles of arbitrary topology. Wang et al. (2015) further advances this field by proposing linear subspace design, unifying linear blend skinning and generalized barycentric coordinates to provide a practical way of controlling deformations.

The representation and editing of objects using Gaussians is a well-explored topic in 3D graphics. In this field, meshes can be modified to simulate Gaussian editing (Guédon & Lepetit, 2024; Huang et al., 2024), or Gaussians can be directly parameterized and manipulated to achieve specific outcomes (Waczyńska et al., 2024b;a). This approach enables flexible and continuous deformations, offering an intuitive method for controlling object shapes and rendering properties, which has proven particularly useful in tasks like texture mapping, surface smoothing, and dynamic simulations.

Gaussians enable precise and flexible editing of objects, providing continuous control over shapes and transformations. Moreover, integrating physics engines enhances these capabilities, allowing for more sophisticated and physically consistent modifications, such as simulating realistic interactions, deformations and movements in 3D environments. (Xie et al., 2024; Borycki et al., 2024).

## 3 MiraGe: Editable 2D Images using Gaussian Splatting

Here, we describe in detail the inner workings of our MiraGe model. We start by presenting classical 3DGS. Next, we present GaMeS-based (Waczyńska et al., 2024b) parametrization of flat Gaussians. In the end, we present our MiraGe and how it relates to prior works.

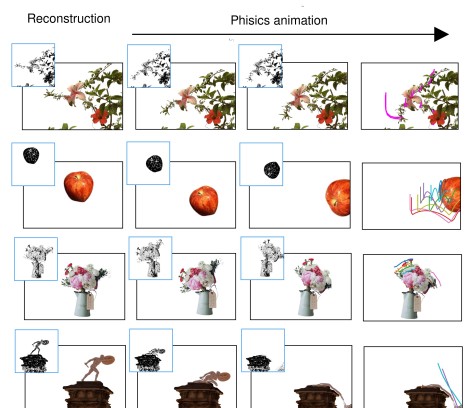

**3D Gaussian Splatting**   3DGS models 3D scene by a set of Gaussian components with color and opacity:

$$\mathcal{G} = \{(\mathcal{N}(\mathbf{m}_i, \Sigma_i), \sigma_i, c_i)\}_{i=1}^p,$$

defined by their mean (position) $\mathbf{m}_i$, covariance matrix $\Sigma_i$, opacity $\sigma_i$, and color $c_i$, which is represented using spherical harmonics (SH) (Fridovich-Keil et al., 2022).

Figure 8: We integrate MiraGe with Material Point Methods (MPM) to achieve realistic alterations of 2D images. The initial column presents the original image, the subsequent two columns display renders captured midway through the simulation, and the final column shows the outcome of the full simulation. The colored lines in the last column trace the paths of randomly chosen points from the simulation.

During the rasterization stage, the 3DGS produces a sorted Gaussian list based on the projected depth information. Then, the $\alpha$-blending method is used to create the image. We refine the Gaussian parameters, color, and opacity in the training phase according to the reconstruction cost function. The optimal number of Gaussians required to represent a given object is not known a priori, and it is non-trivial to adjust the number of Gaussians. Hence, the initial number of Gaussians is a parameter of the method. The authors implement additional strategies for reducing and multiplying Gaussians. Gaussians with low opacity are removed, while those that change rapidly during optimization are multiplied. These strategies make the 3D Gaussian approach very efficient and capable of generating high-quality renders. We used this strategy to reconstruct 2D images, which distinguishes us from GaussianImage.

**GaMeS Parametrization of Gaussian Component**   In MiraGe, we use flat Gaussian components in 3D space. In such a model we use Gaussian components with a covariance matrix $\Sigma$, factored as: $\Sigma = RSSR^T$, where $R$ is the rotation matrix, and $S$ is a diagonal matrix containing the scaling parameters. However, we force one of the scale parameters to be zero. Consequently, we obtain a collection of flat Gaussian:

$$\mathcal{G} = \{(\mathcal{N}(\mathbf{m}_i, R_i, S_i), \sigma_i, c_i)\}_{i=1}^p, \qquad (1)$$

where $S = \text{diag}(s_1, s_2, s_3)$, with $s_1 = \varepsilon$, and $R$ is the rotation matrix defined as $R = [\mathbf{r}_1, \mathbf{r}_2, \mathbf{r}_3]$, with $\mathbf{r}_i \in R^3$. In such a case, we can use GaMeS (Waczyńska et al., 2024b) parametrization to represent flat Gaussian by triangle-face mesh. This mapping is denoted by $\mathcal{T}(\cdot)$. When applied, this parametrization generates a set of triangles labeled as triangle soup.

To outline the GaMeS parameterization, consider a Gaussian component $\mathcal{N}(\mathbf{m}, R, S)$, characterized by the mean $\mathbf{m}$, the rotation matrix $R = [\mathbf{r}_1, \mathbf{r}_2, \mathbf{r}_3]$ and the scaling matrix $S = \text{diag}(\varepsilon, s_2, s_3)$.

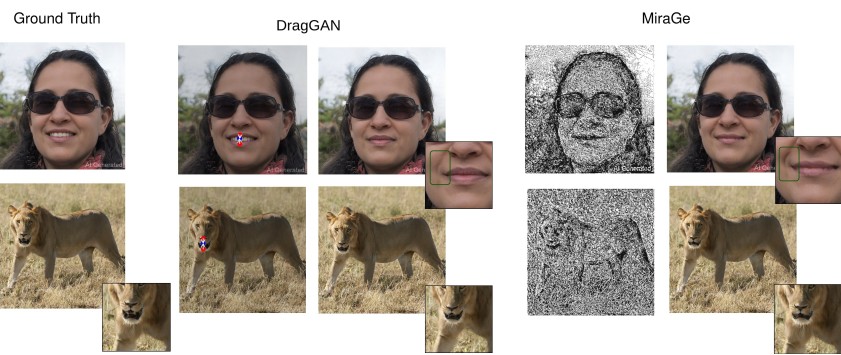

Figure 9: Visual comparison of image editing techniques, demonstrating the effectiveness of representing 2D images with parameterized Gaussians applied to Triangle Soup. This approach enables highly realistic animations, achieving results comparable to those of generative models. Specifically, local editing operations preserve fine details, such as a dimple on a face, without affecting unrelated regions. Moreover, we can achieve precise manipulations, including subtle edits like closing a lion's mouth, underscoring the flexibility and control inherent in our method.

Then its face representation $\mathcal{N}(V)$ is based on a triangle: $V = [\mathbf{v}_1, \mathbf{v}_2, \mathbf{v}_3] = \mathcal{T}(\mathbf{m}, R, S)$ with the vertices defined as: $\mathbf{v}_1 = \mathbf{m}$, $\mathbf{v}_2 = \mathbf{m} + s_2\mathbf{r}_2$, and $\mathbf{v}_3 = \mathbf{m} + s_3\mathbf{r}_3$. Conversely, given a face (triangle) representation $V = [\mathbf{v}_1, \mathbf{v}_2, \mathbf{v}_3]$, we can recover the Gaussian component $\mathcal{N}(\hat{\mathbf{m}}, \hat{R}, \hat{S}) = \mathcal{N}(\mathcal{T}^{-1}(V))$ through the mean $\hat{\mathbf{m}}$, the rotation matrix $\hat{R} = [\hat{\mathbf{r}}_1, \hat{\mathbf{r}}_2, \hat{\mathbf{r}}_3]$, and the scaling matrix $\hat{S} = \text{diag}(\hat{s}_1, \hat{s}_2, \hat{s}_3)$, where the parameters are defined by the following formulas:

$$\hat{\mathbf{m}} = \mathbf{v}_1, \quad \hat{\mathbf{r}}_1 = \frac{(\mathbf{v}_2 - \mathbf{v}_1) \times (\mathbf{v}_3 - \mathbf{v}_1)}{\|(\mathbf{v}_2 - \mathbf{v}_1) \times (\mathbf{v}_3 - \mathbf{v}_1)\|}, \tag{2}$$

$$\hat{\mathbf{r}}_2 = \frac{(\mathbf{v}_2 - \mathbf{v}_1)}{\|(\mathbf{v}_2 - \mathbf{v}_1)\|}, \quad \hat{\mathbf{r}}_3 = \text{orth}(\mathbf{v}_3 - \mathbf{v}_1; \mathbf{r}_1, \mathbf{r}_2), \tag{3}$$

$$s_1 = \varepsilon, \quad \hat{s}_2 = \|\mathbf{v}_2 - \mathbf{v}_1\|, \quad \text{and} \quad \hat{s}_3 = \langle \mathbf{v}_3 - \mathbf{v}_1, \hat{\mathbf{r}}_3 \rangle. \tag{4}$$

Here $\text{orth}(\cdot)$ denotes a single step of the Gram-Schmidt process (Björck, 1994). Accordingly, the corresponding covariance matrix of a Gaussian distribution is given as $\hat{\Sigma} = \hat{R}\hat{S}\hat{S}\hat{R}^T$.

The parametrization enables control over the Gaussians' position, scale, and rotation by manipulating the underlying triangle mesh. Applying transformations to the triangle directly alters the corresponding Gaussian, as illustrated in Fig. 7.

**MiraGe** In this work, we present an approach that leverages the concept of flat Gaussian distributions in 3D space to model a single 2D image as input. Our methodology is grounded in human visual perception. This perspective allows us to reframe the problem: instead of merely processing a pixel matrix, we interpret the images as objects with a fixed spatial configuration in a 3D environment.

We put the 2D image on the $XZ$ plane where the center is situated at axes origin $(0, 0, 0)$ with the fixed distance from the camera origin. In practice, the distance from the plane is a hyper-parameter. In our approach, we model flat objects within 3D space, where the camera distance parameter effectively controls the perceived scale of the object. This relationship allows for intuitive adjustments of object size based on the desired visual effect. For instance, increasing the camera distance can naturally expand the apparent size of background elements like distant mountains (Fig. 4), making it easier to represent them as larger objects without additional modeling complexity. While this feature is beneficial, it is not strictly necessary for most applications.

We propose a method that situates the Gaussians within the $XZ$ plane, ensuring that the entire image remains visible under perspective projection. To achieve this, the possible range of $x$-values and $z$-values is calculated using the camera field of view. We first calculate the deviation from $0$ on the $X$ axis using the similarity of triangles $\text{dev}_z = \text{cam}_{\text{dist}} \cdot \tan(0.5 \cdot \text{Fov}_{\text{vert}})$, where $\text{cam}_{\text{dist}}$ and $\text{Fov}_{\text{vert}}$ are

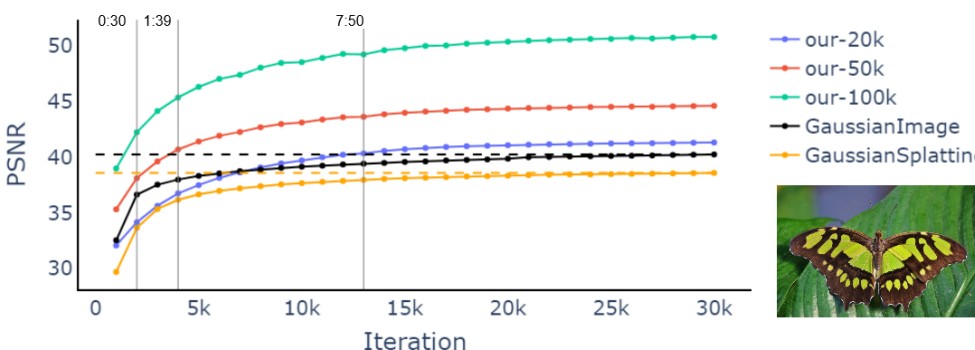

Figure 10: Comparison of PSNR obtained on a butterfly image from DIV2K dataset by MiraGe in comparison with GaussianImage (Zhang et al., 2024) and Gaussian Splatting (Kerbl et al., 2023). The lines with markers represent how the PSNR was changing during training. Different colors represent models trained with different numbers of Gaussians during initialization. The dashed black and orange lines represent the best results obtained during training by GaussianImage and GS, respectively. Vertical lines represent iteration, where MiraGe obtained better results than Gaussian-Image, the time in min:sec format above each line is the training time until this iteration.

camera distance from the $XZ$ plane and camera field of view respectively. The deviation in the $X$ axis can be then computed by multiplying this value by the camera aspect ratio.

Consequently, the initialization of Gaussians is consistently performed on the $XZ$ plane; however, we have opted to permit their movement within the 3D space. Drawing inspiration from three distinct models, we introduce three conceptual approaches for manipulating the spatial positioning of Gaussians.

**Amorphous** The baseline approach how to control Gaussians is based on the classical GaMeS parametrization, initialized randomly on the $XZ$ plane, with the mean parameter's $y$ coordinate set to zero:

$$\mathcal{G} = \{(\mathcal{N}([m_1, 0, m_3], [\mathbf{r}_1, \mathbf{r}_2, \mathbf{r}_3], \text{diag}(\varepsilon, s_2, s_3)), \sigma_i, c_i)\}, \quad (5)$$

where $\mathbf{m} = [m_1, 0, m_3]$ $S = \text{diag}(s_1, s_2, s_3)$, with $s_1 = \varepsilon$, and $R$ is the rotation matrix defined as $R = [\mathbf{r}_1, \mathbf{r}_2, \mathbf{r}_3]$, with $\mathbf{r}_i \in R^3$.

It should be highlighted that we only initialized the Gaussian component on the $XZ$ plane. During training, Gaussians can move amorphously in 3D space. We use the classical loss function $L_1$ combined with a D-SSIM term:

$$\mathcal{L} = (1 - \lambda)\mathcal{L}_1(I, GS(I)) + \lambda\mathcal{L}_{D-SSIM}(I, GS(I)),$$

where $I$ is the input image and $GS(I)$ is the constraint obtained by the Gaussian renderer. While this solution enables the modeling of images using a collection of triangles, often referred to as "triangle soup," it proves insufficient for high-quality representations. During editing, significant artifacts emerge (Fig. 6–Baseline).

**2D** Building on the promising outcomes of GaussianImage, we strategically anchored all Gaussians onto the $XZ$ plane. This configuration allows us to effectively translate the flat image geometry into a spatial framework, bridging perceptual intuition. We set the mean of these components to have zero in the second coordinate. Moreover, we use the projection of flat Gaussians on a 2D plane. Unfortunately, orthogonal projection can produce artifacts. Therefore, we use a rotation of Gaussian components to lay on the $XZ$ plane. Since we use flat Gaussians to extract such rotation we can use a rotation matrix between two vectors to align the vector in 3D (Markley, 1993). We use the notation $\text{Rot}(a, b)$ for the rotation matrix.

MiraGe on 2D plane is define by set of 3G parameterized Gaussian components:

$$\mathcal{G} = \{(\mathcal{N}([m_1, 0, m_3], [\mathbf{r}_1, \mathbf{r}_2, \mathbf{r}_3]\text{Rot}(\mathbf{r}_3, \mathbf{e}_2), \text{diag}(\varepsilon, s_2, s_3)), \sigma_i, c_i)\}, \quad (6)$$

where $\mathbf{e}_2 = [0, 1, 0]$, $\mathbf{m} = [m_1, 0, m_3]$, $S = \text{diag}(s_1, s_2, s_3)$, with $s_1 = \varepsilon$, and $R$ is the rotation matrix defined as $R = [\mathbf{r}_1, \mathbf{r}_2, \mathbf{r}_3]$, with $\mathbf{r}_i \in R^3$.

**Graphite** Unfortunately, 2D-MiraGe produces artifacts when we use modification in 3D space (see the third row in Fig. 6). Such an effect is coursed by the Gaussians, which appear randomly according to the camera position. To solve such a problem and obtain the possibility of 3D modifications, MiraGe allows the Gaussians to leave the $XZ$ plane:

$$\mathcal{G} = \{(\mathcal{N}([m_1, 0, m_3] + \gamma \mathbf{e}_2, [\mathbf{r}_1, \mathbf{r}_2, \mathbf{r}_3] \text{Rot}(\mathbf{r}_3, \mathbf{e}_2), \text{diag}(\varepsilon, s_2, s_3)), \sigma_i, c_i)\}, \tag{7}$$

where $\gamma$ is trainable parameter of translation scale along the vector $\mathbf{e}_2 = [0, 1, 0]$, $\mathbf{m} = [m_1, 0, m_3]$, $S = \text{diag}(s_1, s_2, s_3)$, with $s_1 = \varepsilon$, and $R$ is the rotation matrix defined as: $R = [\mathbf{r}_1, \mathbf{r}_2, \mathbf{r}_3]$, with $\mathbf{r}_i \in R^3$. Such a model allows for the order of Gassians according to camera positions.

By leveraging parameterized Gaussians, we achieved precise manipulation of 2D images directly within their native 2D space, enabling targeted edits of segmented regions and transformations of complete scenes in easier way. While this approach demonstrated substantial promise, we observed significant artifacts when extending manipulations into the 3D domain, particularly along the $Y$-axis, see first and last row in Fig 6.

**Mirror camera** We employ a novel approach utilizing two opposing cameras positioned along the $Y$ axis, symmetrically aligned around the origin and directed towards one another. The first camera is tasked with reconstructing the original image, while the second models the mirror reflection. We introduced the mirror camera to ensure that Gaussians remain confined within a specific spatial region between the cameras, enhancing control and precision. The reflection can be effectively represented by horizontally flipping the image, denoted as $\mathcal{M}(I)$. This mirror-camera setup enhances the fidelity of the generated reflections, providing a robust solution for accurately capturing visual elements. We consider the additional camera as a means of augmenting the dataset to improve the accuracy of the representation. The MiraGe is initialized according to equation Eqn. 5 and utilizes a cost function: $\mathcal{L}(I) + L(\mathcal{M}(I))$. We simultaneously model both the image and its mirrored reflection, as shown in the second row in Fig. 6. We provide numerical comparison in the ablation study in the Appendix.

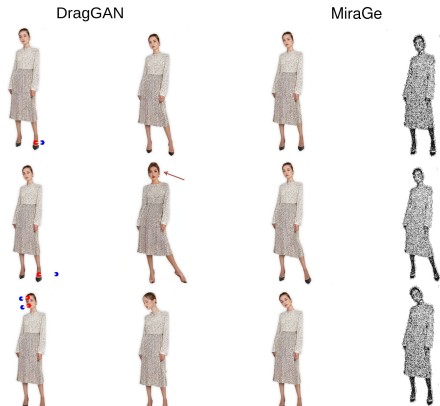

Figure 11: We compare the animation capabilities of MiraGe with those of the DragGAN model, highlighting the advantages of our Gaussian-based image representation. This approach enables highly realistic edits by not relying on generative techniques. Our method offers greater control during animation. For example, adjusting the position of a leg does not inadvertently alter facial features.

After thorough experimentation, we find that our model, Amorphous-MiraGe, utilizing a mirror camera, achieves state-of-the-art reconstruction results. This model demonstrates significant advantages over alternative methods in terms of both performance and outcome quality.

**Editability** The ability to manipulate Gaussians based on their spatial positioning empowers MiraGe to effectively edit 2D images. When utilizing a mirror camera, the quality of the resulting images is sufficiently high, enabling the parameterization and animation of Gaussians to significantly reduce artifacts. Our findings demonstrate that our model facilitates the animation of both segmented objects and entire scenes. Users can create manual animation, or leverage automated processes using physics engines like Taichi_elements or Blender (Fig. 3,8). To incorporate MiraGe with the 2D physics engine, we use 2D-MiraGe (see, Fig. 8). In Fig. 9, we demonstrate that our method can also be applied to edit more complex scenes, such as changing human expression.

We argue that the Graphite-inspired model allows the creation of attractive compositions made of multiple images that effectively present the positive attributes of the layered structure, like Graphite, through the strategic positioning of Gaussians.

## 4 EXPERIMENTS

We split the experimental section of our paper into two main parts. First, we demonstrate that our approach achieves high-quality 2D reconstruction by comparing it with existing models. Second, we highlight the versatility of our method in image editing both full scenes (Fig. 9) and selected objects (Fig. 1, 7), presenting examples of user-driven modifications and demonstrations involving physical simulations (Fig. 3, 8).

Table 1: Quantitative comparison with various baselines in PSNR and MS-SSIM. MiraGe gives state of the art results. Model-$x$ denotes that the model was initialized with $x$ Gaussians.

|  | Kodak dataset | | DIV2K dataset | |
|---|---|---|---|---|
|  | PSNR ↑ | MS-SSIM ↑ | PSNR ↑ | MS-SSIM ↑ |
| WIRE | 41.47 | 0.9939 | 35.64 | 0.9511 |
| SIREN | 40.83 | 0.9960 | 39.08 | 0.9958 |
| I-NGP | 43.88 | 0.9976 | 37.06 | 0.9950 |
| NeuRBF | 43.78 | 0.9964 | 38.60 | 0.9913 |
| 3DGS | 43.69 | 0.9991 | 39.36 | 0.9979 |
| GaussianImage-70k | 44.08 | 0.9985 | 39.53 | 0.9975 |
| GaussianImage-100k* | 38.93 | 0.9948 | 41.48 | 0.9981 |
| MiraGe-70k (our) | 57.41 | 0.9998 | 53.22 | 0.9996 |
| MiraGe-100k (our) | 59.52 | 0.9999 | 54.54 | 0.9998 |

**Reconstruction quality**  Our image reconstruction assessment utilizes two widely-recognized datasets. Specifically, we employ the Kodak dataset[3], which includes 24 images at a resolution of $768 \times 512$, alongside the DIV2K validation set (Agustsson & Timofte, 2017), which involves $2\times$ bicubic downscaling and comprises 100 images with sizes ranging from $408 \times 1020$ to $1020 \times 1020$. The dataset was selected to facilitate direct comparison with the work of GaussianImage. As a baselines we use competitive INR methods GaussianImage (Zhang et al., 2024), SIREN (Sitzmann et al., 2020b), WIRE (Saragadam et al., 2023b), I-NGP (Müller et al., 2022), and NeuRBF (Chen et al., 2023).

In Tab. 1, we demonstrate the performance outcomes of different methods on the Kodak and DIV2K datasets. We see that our proposition outperforms the previous solutions on both datasets. The quality measured by both metrics shows significant improvement compared to all the previous approaches. Fig. 10 illustrates a general trend observed during training in the contest of image reconstruction. The selection of hyperparameters, including the number of iterations, was inspired by the principles of 3DGS. We provide ablation studies and extensive numerical analyses in the appendix for further insights.

It is important to note that although our model takes longer to train, it quickly achieves better results than

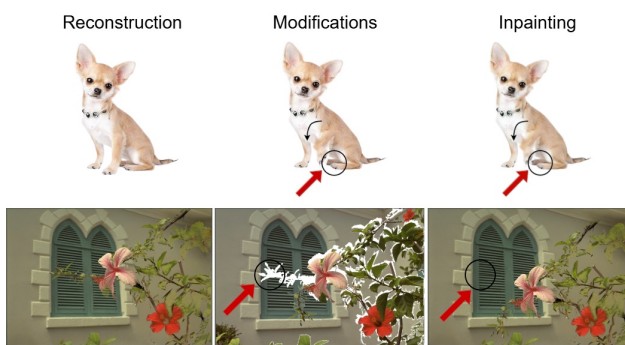

Figure 12: MiraGe model allows modifications in 3D space, but the model is limited by 2D images, which was used in training. When we move some elements from the foreground, we cannot see the background since the model only reconstructs objects. Next, we can use image Inpainting to fill the missing parts, allowing for more realistic modifications.

GaussianImage. The trend we observe is illustrated in Fig. 10, which also includes the number of initial Gaussians, indicating how densely the space has been filled. We see a clear upward trend in performance as the density of the Gaussian initialization increases.

**Manual modification**  MiraGe allows for manual manipulation of 2D images. By leveraging GaMeS parameterization, each Gaussian component is represented as a triangle. Vertices can then be independently adjusted and moved within 3D space, enabling flexible image modification (Fig. 2, 7).

We demonstrate examples of modifications using datasets such as DIV2K, Kodak, and Animals[4]. Additionally, we generated our own 2D images using DALL-E 3 to illustrate the benefits of our method. We can obtain modifications of small details like changing fingers' position (Fig. 1), human

---

[3] https://r0k.us/graphics/kodak/
[4] https://www.kaggle.com/datasets/alessiocorrado99/animals10

facial expressions (Fig. 16) or dog poses (Fig. 12). As MiraGe can trained in a 3D context, we can implement modifications in the third dimension to create the illusion of 3D transformation (Fig. 3, 6).

It is crucial to note that when we displace elements from the foreground, the background remains unseen because the model only reconstructs the objects. This is demonstrated in Fig. 12, where artifacts are apparent on the hind paw of the depicted animal. To reduce such problem we can use Inpainting (Perche-Mahlow et al., 2024) on the image background.

We conducted a comparative analysis of our editing approach against the DragGAN model (Pan et al., 2023). Here, we focused on the ability to perform localized edits, such as closing the mouth, while preserving other features, such as dimples (see Fig. 9). Visual results, presented in Fig. 11, highlight key distinctions between the two models. As DragGAN is a generative model, modifications often result in unintended global transformations, for instance, attempting to adjust a leg's position may inadvertently modify facial features. In contrast, our method demonstrates the capability to move elements like the leg with realistic results and without compromising other aspects of the image.

**Physics application in MiraGe** Using 2D-MiraGe we can express Gaussian components with a 2D point cloud. Therefore, we can use MPM (Hu et al., 2018) based physics engine

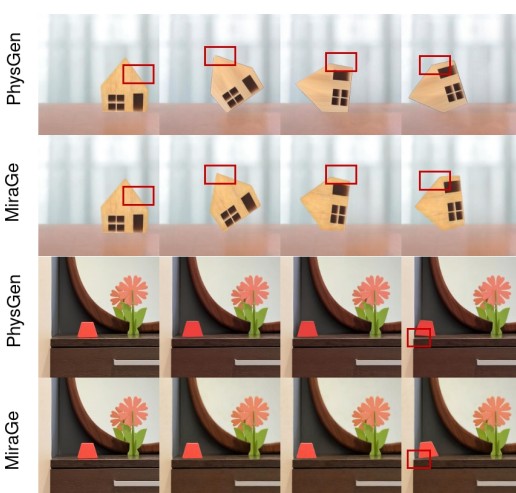

Figure 13: Comparison between PhysGen (Liu et al., 2025) and MiraGe. Our render has properly solved the issue with the house borders. Moreover, PhysGen has slightly changed the shape of the building (e.g., the elongated house tip). In another animation, reasonable doubts related to the correctness of the simulated physics of PhysGen arise when a reader follows the behavior of a red element upon hitting the wall; contrary to everyday experience, the front part (instead of the back part) of the figure bounces off the tabletop.

implemented, for example, in Taichi_elements. This high-performance physics engine supports multiple materials, including elastic objects and sand. We use inspiration from GASP (Borycki et al., 2024) and train simulation on 2D points then use physical deformation on triangle soup. In Fig. 8, we present simulation results obtained using Taichi_elements. As we can see, we can add physical properties to 2D objects. On the other hand, using Amorphous-MiraGe or Graphite-MiraGe we can use Blender and modify directly parameterized flat 3D Gaussian (Fig. 3). Moreover, we compare MiraGe with PhysGen (Liu et al., 2025) (Fig. 13). Our renderer has successfully resolved the issue with the house borders. Additionally, PhysGen has slightly altered the shape of the building. Furthermore, in another animation, questions arise about the accuracy of PhysGen's simulated physics. Specifically, when observing the behavior of a red element striking a wall, the outcome contradicts everyday experience, i.e., instead of the back part of the figure bouncing off the tabletop, the front part rebounds (Fig. 13).

## 5 CONCLUSION

In this paper, we introduce MiraGe that uses flat 3D Gaussian components to model 2D images. MiraGe gives state-of-the-art reconstruction quality and simultaneously allows image manipulation. Furthermore, we can modify photos on a plane (Fig. 8) and in 3D space (Fig. 3). In consequence, we obtain the illusion of 3D-based modifications. Furthermore, we can combine our solution with a physics engine to obtain realistic motion in the image. Conducted experiments show that MiraGe is applicable in many different scenarios and produces high-quality simulations. **Limitation** It is crucial to note that the model is not generative, so improper adjustment of Gaussian positions can cause gaps in the image (e.g. a missing dog's paw Fig. 12). While the model can produce realistic changes, significant modification may introduce visual artifact. Moreover, our model requires encoding more parameters than GaussianImage to achieve high-quality image reconstruction for animation. Addressing this trade-off will be a focus of our future work.

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

# 6 APPENDIX

Here, we provide a comprehensive overview of the implementation details. Furthermore, we present supplementary experimental results, such as extended performance evaluations and ablation studies focusing on camera settings.

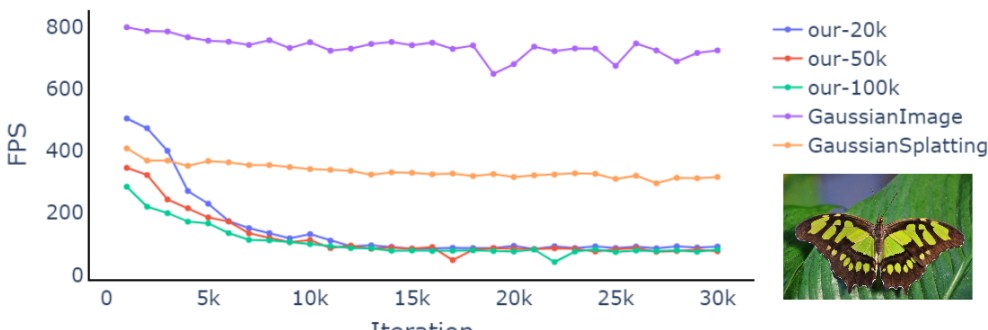

Figure 14: Comparison of FPS obtained on a butterfly image from DIV2K dataset by MiraGe in comparison with GaussianImage (Zhang et al., 2024) and Gaussian Splatting (Kerbl et al., 2023). The experiment was performed on the rtx 4070 GPU.

## 6.1 IMPLEMENTATION DETAILS

The source code for this project will be made publicly available on GitHub. Our code was developed based on the GaMeS framework, and it is distributed under the GS Vanilla license. Computational experiments in main paper were conducted using NVIDIA GeForce RTX 4070 Laptop version and NVIDIA GeForce RTX 2080. Appendix time comparisons were reported using NVIDIA GeForce RTX 2080.

Building upon the GaMeS framework, we initialized the Gaussian distributions to lie perpendicular to the $XZ$ plane. In our model, where all Gaussians are constrained to a 2D plane at rendering time, we consider only the rotation angle, denoted as $\phi$, as the primary rotation parameter. To facilitate the rendering of Gaussians positioned on the $XZ$ plane, $\phi$ serves as the primary learning parameter. The corresponding quaternions of rotation are computed as follows: for rotation about the x-axis $q_x = [cos(\frac{\phi}{2}), sin(\frac{\phi}{2}), 0, 0]$, and for the z-axis $q_z = [cos(\frac{\pi}{2}), 0, 0, sin(\frac{\pi}{2})]$. Since no rotation occurs about the y-axis, the quaternion remains $q_y = [1, 0, 0, 0]$. These quaternions are then combined through multiplication to form a new rotation matrix, ensuring precise alignment of the Gaussians on the $XZ$ plane.

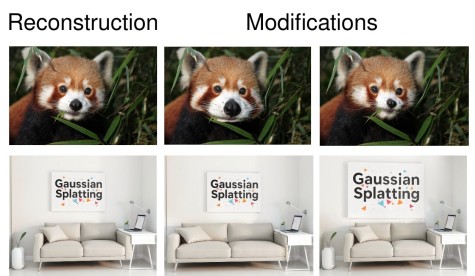

Figure 15: MiraGe enables modifying 2D images, such as adjusting the scene's elements' sizes.

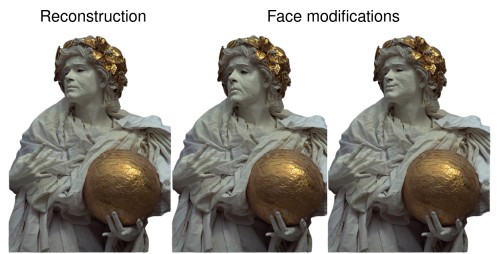

Figure 16: MiraGe allows us to produce realistic modifications of small details like changing human facial expressions.

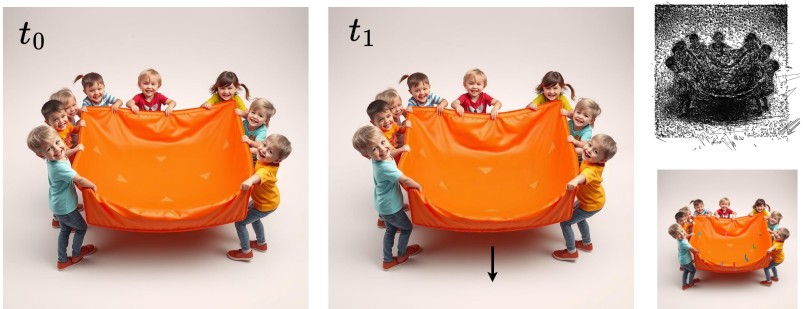

Figure 17: MiraGe allows for manual image edits and for using a physics engine for real-life-like image modifications.The left image illustrates a Gaussian representation achieved through a triangle mesh triangle soup, while the accompanying point-based depiction provides finer details, offering a more refined visual comparison.

Figure 18: MiraGe allows for the modification of larger scenes. We can selectively alter specific areas and introduce smooth movements or material adjustments. In this example, the bottom of the blanket is shown in motion. This, along with other modifications, is available in the supplementary files as videos.

Table 2: Ablation study of the effect of adding the mirror camera as augmentation technique on training time and the output image quality measured in widely recognized metrics: PSNR, MS-SSIM, LPSIS. Experiment was performed with initial 100k Gaussians.

| Kodak dataset | | | | | |
|---|---|---|---|---|---|
| Gaussian control method | Camera Setting | PSNR ↑ | MS-SSIM ↑ | LPSIS ↓ | Training Time(s) ↓ |
| Amorphous | One camera | 51.56 | 0.9996 | 0.0050 | 448.73 |
| | Mirror cameras | 59.52 | 0.9999 | 0.0005 | 639.66 |
| Graphite | One camera | 42.49 | 0.9948 | 0.2984 | 398.54 |
| | Mirror cameras | 46.90 | 0.9983 | 0.1238 | 739.66 |
| 2D | One camera | 42.75 | 0.9950 | 0.2931 | 552.80 |
| | Mirror cameras | 48.82 | 0.9987 | 0.0071 | 942.78 |
| DIV2K dataset | | | | | |
| Gaussian control method | Camera Setting | PSNR ↑ | MS-SSIM ↑ | LPSIS ↓ | Training Time(s) ↓ |
| Amorphous | One camera | 46.00 | 0.9991 | 0.0162 | 690.98 |
| | Mirror cameras | 54.54 | 0.9998 | 0.0033 | 946.35 |
| Graphite | One camera | 40.02 | 0.9949 | 0.0312 | 582.50 |
| | Mirror cameras | 46.52 | 0.9986 | 0.0117 | 1082.41 |
| 2D | One camera | 39.99 | 0.9949 | 0.0310 | 869.62 |
| | Mirror cameras | 46.32 | 0.9985 | 0.0124 | 1278.33 |

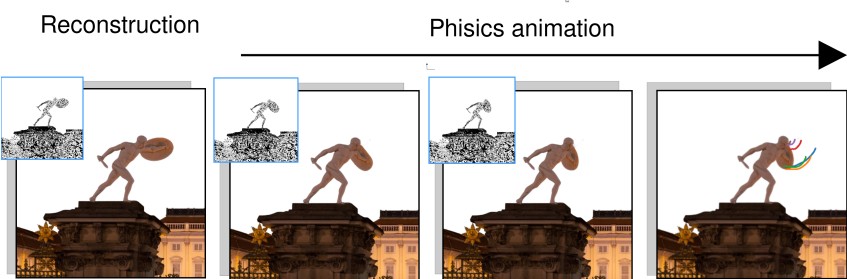

Figure 19: MiraGe can be integrated with Blender, by using flat 3D Gaussians in 3D space. The initial column presents the original image, the subsequent two columns display renders captured midway through the simulation, and the final column shows the outcome at the simulation's conclusion. The colored lines in the last column trace the paths of 10 randomly chosen points from the simulation.

## 6.2 SUPPLEMENTARY NUMERICAL FINDINGS FROM THE PRIMARY PAPER

We conducted an extensive analysis of the MiraGe model due to its unique ability to control the behavior of Gaussians. Three distinct settings for Gaussian movement were explored:

- Amorphous the first allows Gaussians to move freely in 3D space,

- 2D: the second restricts their movement to align parallel to the $XZ$ plane

- Graphite the third confines all Gaussians to the $XZ$ plane, effectively creating a 3D representation.

A qualitative analysis was performed, considering the impact of the mirror camera (see Tab. 2), as well as the effect of varying the number of initial Gaussians on the overall model behavior (see Tab. 3). We also examined the impact of the camera using Frames Per Second (FPS) metric and storage memory (see Tab. 4). Given the ongoing development of various 3D Gaussian Splatting compression techniques, we employed the .spz[5] tool to effectively compress the data.

Due to our particular focus on animation, we analyzed FPS trends to benchmark real-time performance. Fig. 14 shows that while our model introduces a higher number of parameters, leading to a decrease in FPS compared to GaussianImage, it maintains the ability to render animations in real time.

Tab. 2 shows the mirror camera view as augmentation technique improves significantly the representation's fidelity of every proposed Gaussian method. This behavior can be detected with the help of any of the measured metrics, i.e., PSNR, MS-SSIM and LPSIS. The drawback of improving the image quality is a longer training time required. The ablation study presented in Tab. 3 similarly suggests that our model scales well with the number of Gaussians used during model initialization. The striking example here is an average 62.12 PSNR score achieved by Amorphous method on Kodak dataset. The price paid in time of training grows here slower, i.e., increasing the number of starting Gaussians by an order of magnitude results in more extended though comparable training period length.

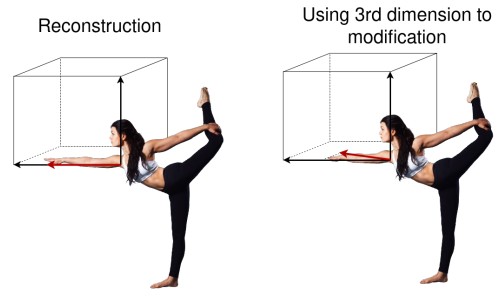

Figure 20: MiraGe simplifies intuitive editing of image, allowing transformations such as adjusting the tilt of a hand with minimal complexity. This is achieved by modifying the object along the third dimension.

---

[5] https://github.com/nianticlabs/spz

## 6.3 EXTENSION OF EXAMPLES MODIFICATION AND ARTIFACTS

Animating a full scene can be non-trivial, but it is possible. Fig. 15 demonstrates how a painting can be enlarged to visualize the impact of its placement in a room, offering a clear view of the potential arrangement. It is also possible to animate small, localized areas of the image, as demonstrated in Fig. 16. For the facial animation, we utilized the Lattice modifier in Blender. MiraGe enables manual image editing and incorporates a physics engine for image modifications (Fig. 17, 18, 19). It is crucial to remember that if certain Gaussians are shifted without considering their dependencies on others, the image will be disrupted. Therefore, the relationships between the Gaussians must be carefully modeled. We demonstrate this concept with the example of children playing with a blanket (Fig. 18). Despite the movement of the blanket (as seen in the supplementary video), the image remains uninterrupted and coherent.

A simple editing concept using 3D is shown in Fig 20. Fig. 19 illustrates a sculpture where the movement of the hand is achieved by adjusting the position of the shield behind the warrior. The image representation, based on parameterized Gaussians, facilitates precise editing of fine details within the 3D space.

Figure 21: Example of artifacts generated during animation, typically due to imperfect rendering. The model was trained on a white background, leaving residual white Gaussians along the border of the camel's muzzle, leading to artifacts. In this instance, the Graphine-MiraGe performed best in handling the head-turning movement

Integrating the representation into Blender can introduce automatic adjustments that may result in visual artifacts (Fig. 21), particularly when training on images with a white background. These modifications can lead to unrealistic renderings that are challenging to detect through automated means and currently require subjective evaluation by a human observer.

## 6.4 SOCIAL IMPACT

The model can be applied to generate novel image transformations used to dataset augmentation, facial recognition (Sanil et al., 2023), distortion correction, and visualizing architectural designs (Gerstweiler et al., 2018), as shown in Fig. 15. In the field of medical imaging, it is utilized for refining anatomical models and enhancing the accuracy of surgical simulations and diagnostic tools (Lin et al., 2023). Additionally, in 2D computer games, the model facilitates more realistic animations by incorporating physics-based effects (Mohd et al., 2023).

Table 3: Measuring the influence of the initial number of Gaussians on the quality of the Image reconstruction. Experiment was performed using mirror camera view for every table entry.

| Kodak dataset | | | | | |
|---|---|---|---|---|---|
| Gaussian control method | Initial Gaussians | PSNR ↑ | MS-SSIM ↑ | LPSIS ↓ | Training Time(s) ↓ |
| Amorphous | 10k | 50.66 | 0.9987 | 0.3531 | 584.84 |
| | 50k | 55.54 | 0.9997 | 0.0033 | 634.65 |
| | 100k | 59.52 | 0.9999 | 0.0005 | 639.66 |
| | 150k | 62.12 | 0.9999 | 0.0002 | 676.10 |
| Graphite | 10k | 40.39 | 0.9940 | 0.0599 | 651.32 |
| | 50k | 44.90 | 0.9973 | 0.2024 | 732.91 |
| | 100k | 46.90 | 0.9983 | 0.1238 | 739.66 |
| | 150k | 48.16 | 0.9987 | 0.0105 | 801.18 |
| 2D | 10k | 39.75 | 0.9886 | 0.0769 | 857.30 |
| | 50k | 45.03 | 0.9955 | 0.2789 | 876.56 |
| | 100k | 48.82 | 0.9987 | 0.0071 | 942.78 |
| | 150k | 50.54 | 0.9992 | 0.0031 | 955.86 |
| DIV2K dataset | | | | | |
| Gaussian control method | Initial Gaussians | PSNR ↑ | MS-SSIM ↑ | LPSIS ↓ | Training Time(s) ↓ |
| Amorphous | 10k | 49.53 | 0.9987 | 0.0322 | 852.19 |
| | 50k | 52.23 | 0.9995 | 0.0112 | 902.80 |
| | 100k | 54.54 | 0.9998 | 0.0033 | 946.35 |
| | 150k | 56.40 | 0.9999 | 0.0014 | 975.44 |
| Graphite | 10k | 40.75 | 0.9959 | 0.0457 | 983.41 |
| | 50k | 44.67 | 0.9980 | 0.0216 | 1008.52 |
| | 100k | 46.52 | 0.9986 | 0.0117 | 1082.41 |
| | 150k | 47.61 | 0.9989 | 0.0083 | 1103.69 |
| 2D | 10k | 38.40 | 0.9920 | 0.0616 | 1166.09 |
| | 50k | 42.86 | 0.9967 | 0.0275 | 1256.62 |
| | 100k | 46.32 | 0.9985 | 0.0124 | 1278.33 |
| | 150k | 48.46 | 0.9990 | 0.0065 | 1415.54 |

Table 4: Ablation study of the effect of adding the mirror camera as augmentation technique on Kodak dataset measured using Frames Per Second (FPS) and memory storage.

| Kodak dataset | | | | |
|---|---|---|---|---|
| Gaussian control method | Camera Setting | FPS | Memory (MB) | Compressed memory (MB) |
| Amorphous | One camera | 583.28 | 31.25 | 2.42 |
| | Mirror cameras | 620.10 | 117.25 | 7.80 |
| Graphite | One camera | 1157.75 | 30.71 | 2.68 |
| | Mirror cameras | 650.75 | 117.91 | 9.22 |
| 2D | One camera | 1130.08 | 30.69 | 2.68 |
| | Mirror cameras | 418.39 | 173.64 | 12.82 |

