# OpenReview forum: "MiraGe: Editable 2D Images using Gaussian Splatting"
_ICLR.cc/2025/Conference — Submitted to ICLR 2025_

### Official Review · Reviewer_9FhM · 2024-10-25

**Soundness:** 2
**Presentation:** 2
**Contribution:** 2
**Rating:** 5
**Confidence:** 5

**Summary:**

This paper introduces a novel method, MiraGe, which uses mirror reflections to perceive 2D images in 3D space and employs flat-controlled Gaussians for precise 2D image editing.

**Strengths:**

The proposed method achieving state-of-the-art reconstruction quality of 2D image using Gaussian Splatting, also enables the manipulation of 2D images within 3D space, creating the 3D effects.

**Weaknesses:**

1. Lack of understanding of 3D structure: This paper elevates 2D image editing to 3D space, but it overlooks the importance of capturing the underlying 3D structure in the 2D image. As a result, the proposed method is limited in its ability to handle editing tasks that involve invisible parts or complex transformations. For example, while the method can achieve editing by dragging existing parts, it fails to handle editing tasks that require manipulating hidden parts, such as the hidden leg of a cat (as shown in Figure 2). Additionally, most of the manipulation effects are limited to affine transformations or local rigid editing, which can also be accomplished using standard 2D editing tools like Photoshop. Without a robust representation of the 3D structure, it is difficult to guarantee that the complex editing capabilities offered by the proposed method in 3D space will always translate to reasonable results when reflected in the 2D image.

2. The authors should provide a more detailed explanation of why the mirror camera is necessary for the proposed method and how it contributes to the overall performance of the method. Additionally, the comparison between the second row (Amorphous + mirror camera) and the fourth row (Graphic + mirror camera) in Figure 6 suggests that they achieve almost the same effects, which does not clearly illustrate the necessity of Graphite.

3. I think the paper is hard to follow for me, which take me a while. I suggest the authors to improve the readability of the paper further. The paper layout and formatting is a little messy, and the writing style is not very clear. The authors should consider revising the paper to make it more accessible to readers. And the experimental results are not very convincing and has insufficient experimental length. Concretely, this paper lack the visual comparison with baseline methods and 2D editing methods.

**Questions:**

1. Questions: I don't queit get the necessity of intriducing the mirror camera. Could you provide more detailed explanation on this?

2. Suggestions: It is recommended that the authors show more convincing editing visual results that could prove that your method has a good understanding of underlying 3D structure. By the way, I think the paper is hard to follow for me, which take me a while. I suggest the authors to improve the readability of the paper.

---

### Official Review · Reviewer_g8tX · 2024-10-27

**Soundness:** 2
**Presentation:** 2
**Contribution:** 2
**Rating:** 5
**Confidence:** 3

**Summary:**

The paper aims to represent 2D images with high reconstruction quality in an editable fashion. To this end, it positions 2D images in 3D space and manipulates them accordingly. More specifically, the paper integrates GaussianImage and GaMes parameterization, along with the proposed mirror loss.

**Strengths:**

The authors provide numerous figures and videos for visualization, and the reconstruction performance is significant.

**Weaknesses:**

First of all, although not significant, the paper has been submitted in the camera-ready format rather than the under-review format. Therefore, I apologize for not being able to pinpoint the exact locations of each word and sentence.

It would be highly beneficial for the paper to provide more concrete reasons for why this low-level “manual editing” is an important issue, as manually editing Gaussians is quite trivial. Furthermore, there is an inconsistency regarding this matter. In the introduction, it states that GaussianImage lacks the capability for manual editing, but in the related works section, it contradicts this by claiming that Gaussians enable precise and flexible editing. More detailed explanations of the limitations of GaussianImage in manual editing, as well as a clearer highlighting of the strengths of this paper, would better clarify its intentions.

Additionally, more thorough and extensive experiments could be conducted, as Table 1 alone is insufficient to demonstrate how much each component affects performance. Specifically, an ablation study on GaMes parameterization and mirror loss concerning representation performance would clearly demonstrate the role of each component. A quantitative comparison between Amorphous, 2D, and Graphite on various metrics would help readers understand the benefits of each method more clearly. Moreover, quantitative and qualitative results on other manual editing methods would further enhance the paper.

(Minor) Additionally, there are some minor misused or misleading words and phrases that could be easily corrected without disrupting the main storyline. For example, “natural networks” in the Related Works section seems to refer to neural networks. The phrase “Training the Gaussian component is challenging” in Section 3 could be more clearly articulated. A clearer version might state like "the optimal number of Gaussians required to represent a given input (image) is not known a priori, and it is non-trivial to adjust the number of Gaussians.

**Questions:**

As mentioned in the weaknesses section, I would appreciate it if the authors could provide more persuasive motivation and clearly outline the contributions of the proposed method.

---

### Official Review · Reviewer_RT9X · 2024-11-03

**Soundness:** 4
**Presentation:** 4
**Contribution:** 4
**Rating:** 8
**Confidence:** 4

**Summary:**

MIRAGE proposes to represent images using flat Gaussians by using the notion of mirror reflections to create 3D awareness. The method achieves good reconstruction and editing of images in 3D space. The authors propse 3 slightly different techniques for controlling Gaussians- Amorphous, 2D, and Graphite. The authors evaluate the approaches on the Kodak and Div2 dataset.

**Strengths:**

* The paper is well written and neatly presented. Both the problem and the solution are well-motivated.
* It explores an interesting direction to edit images in 3D using flat Gaussians, compared to exisitng generative based methods.
* The paper explores different approaches for controlling Gaussians- Amorphous, 2D, and Graphite. The formulation for Graphite inspired from mirror reflections is quite interesting.
* Evaluation setup seems sound and the authors include an ablation as well. The mirror cameras show a good improvement in scores.
* The method performs very well on reconstruction and the editing results look promising while keeping the original structure in place as it's not generative
* The implementation is efficient by leveraging physcis engines such as Taichi

**Weaknesses:**

* Manipulating Gaussians directly to perform edits seems quite tedious and even challenging for users (compared to instruction based editing in InstructPix2Pix for example)
* While the method is promising, the edits performed have small delta with the original as it's mostly with scaling and shifting Gaussians. This limitation is a natural con of lacking a generative prior.
* The time taken to train the Gaussians on a single image is rather long 450~950 seconds.

**Questions:**

* Can the authors describe in more details on how the exact editing was performed for the different results? Eg: facial expression change, scaling the painting/mouth

---

### Official Review · Reviewer_Jt4G · 2024-11-04

**Soundness:** 3
**Presentation:** 3
**Contribution:** 3
**Rating:** 6
**Confidence:** 3

**Summary:**

This paper combines GaMeS framework (which maintains a mesh representation of a 3DGS such that mesh edits propagate to the GS), with the idea of GaussianImage (which uses GS to represent 2D images). They analyze and fix several issues that arise, including moving from 2D to 3D gaussians to allow for 3D editing. An additional point of novelty is that both the image and its mirror are optimized when training the GS, improving quality dramatically. The authors present qualitative results showing complex editing of 2D images enabled by their method.

**Strengths:**

*I think this is a novel and creative combination of existing methods, and the additional “mirror” loss acts as an effective regularizer

*Evaluation of reconstruction quality is sound and strongly favors the proposed approach

**Weaknesses:**

*I did not see training/inference time and memory in the quantitative evaluation table

*The editing capability does not have a comparative evaluation. Because this is a claimed contribution, I think the authors should find baselines and demonstrate why the proposed approach is superior for interactive image editing. Some of the following references could be considered in the evaluation:

[1] Pan, Xingang, et al. "Drag your gan: Interactive point-based manipulation on the generative image manifold." ACM SIGGRAPH 2023 Conference Proceedings. 2023.
[2] Shi, Yujun, et al. "Dragdiffusion: Harnessing diffusion models for interactive point-based image editing." Proceedings of the IEEE/CVF Conference on Computer Vision and Pattern Recognition. 2024.
[3] Jacobson, Alec, et al. "Bounded biharmonic weights for real-time deformation." ACM Trans. Graph. 30.4 (2011): 78.
[4] Wang, Yu, et al. "Linear subspace design for real-time shape deformation." ACM Transactions on Graphics (TOG) 34.4 (2015): 1-11.


*Visualizations of the mesh are not shown when editing, and how the mesh edit propagates to the Gaussian splat. As an ablation, it would help to show why we can’t live with the edited (textured) mesh and need to map the edits to the Gaussian Splat.

*Evaluation of reconstruction quality is missing comparative qualitative examples. Because the PSNRs/SSIMs are generally quite high, it may be tough to appreciate these improvements qualitatively.

**Questions:**

The selection process when editing is not shown (e.g. what tools/algorithms are used to select patches of vertices to move around?)

Fig. 12 needs more context and intuition as to why the FPS drops after training for longer for the proposed method, but not for Gaussian Image/GS?

---

### Comment · Area_Chair_ZFjH · 2024-11-25
**Last day for interactive discussions!**

Dear authors and reviewers,

The interactive discussion phase will end in one day (November 26). Please read the authors' responses and the reviewers' feedback carefully and exchange your thoughts at your earliest convenience. This would be your last chance to be able to clarify any potential confusion.

Thank you,
ICLR 2025 AC

---

### Author Response · Authors · 2024-12-01
**Global response**

We would like to sincerely thank all the Reviewers for appreciating our work and for their comments and reviews. We believe their feedback has greatly enhanced the quality of our research.

To summarize our efforts in addressing the reviewers’ feedback We have included additional comparative experiments with preceding methods, such as DragGAN and PhysGen, providing a broader context for our work. Regarding the editing capabilities, we have demonstrated the use of simple background inpainting methods that can be effectively integrated into the proposed pipeline (Fig. 12). Furthermore, we have added a few supplementary videos (in mp4 format) illustrating physical experiments:
- Displacement of a car and a rolling wooden house block using Taichi_Elements.
- Simulation of a domino effect utilizing Blender's Rigid body modifier.

We hope these additions provide clarity and strengthen the manuscript. If there are any additional questions or suggestions, please do not hesitate to post them and we would be glad to provide further clarification.

---

### Meta-Review · Area_Chair_ZFjH · 2024-12-20

**Metareview:**

The submission received mixed reviews. The reviewers appreciate the innovation of using flat Gaussians for 3D-like editing while demonstrating strong reconstruction results. The major concerns from g8tX and 9FhM were on the accuracy of claims on 3D editability and physical simulation-readiness, as well as the practical necessity of Gaussian representations for image editing. The AC carefully read through the paper, the reviewers' comments, the authors' rebuttal and the discussions. While the paper presented an interesting approach with well-executed experiments, the AC agrees with g8tX and 9FhM that more extensive discussions on the practicality and enabled capabilities should be provided to better justify the contributions. The AC also believes that the work has deviated away from the domain of learning representations / neural networks / AI, and the manuscript may be more suitable in computer graphics or vision conferences (e.g. SIGGRAPH or CVPR). As such, the AC recommends rejection at this time.

**Additional Comments On Reviewer Discussion:**

The reviewers raised questions mostly regarding lack of comparisons for editing (Jt4G, g8tX, 9FhM), missing visualizations (Jt4G, g8tX, 9FhM), and the positioning / claims on 3D editability and physics simulability (g8tX, 9FhM). The questions on experiments and visualizations were adequately addressed by the authors, but g8tX and 9FhM were not convinced on the claims and positioning of the paper, leading to doubts on the contributions.

---

### Decision · Program_Chairs · 2025-01-22

Reject